# Development of the Protocol of the Occupational Risk Assessment Method for Construction Works: Level of Preventive Action

**DOI:** 10.3390/ijerph17176369

**Published:** 2020-09-01

**Authors:** Antonio José Carpio-de los Pinos, María de las Nieves González-García

**Affiliations:** 1Department of Applied Mechanics and Project Engineering, School of Aerospace and Industrial Engineering of Toledo, University of Castilla-La Mancha, 45071 Toledo, Spain; AntonioJose.Carpio@uclm.es; 2Departamento de Construcciones Arquitectónicas y su Control, Escuela Técnica Superior de Edificación, Universidad Politécnica de Madrid, 28040 Madrid, Spain

**Keywords:** health and safety, risk assessment, construction, place of work, assessment protocol

## Abstract

Risk assessment on a construction site is based on the probability and consequences of the accident. But due to the complexity of the construction processes, this new methodology for the evaluation of occupational risks, called Level of Preventive Action, develops a new parameter for evaluating preventive action based on documentary environment that reflects the complexity of the work units, location and interdependence, construction environment referred to construction and protection systems, and social environment relative to the perception of the environment and the workers’ emotional states. The evaluation criteria of the new method are established by developing the William T. Fine methodology and incorporating concepts such as risk tolerance, the importance of work and personal satisfaction, which justify the degree of correction of preventive actions. This methodology determines the amount of preventive action control that is required during the construction process. This research proposes a risk assessment protocol adapted to construction sites based on specialized technical observation with a psychosocial survey on site. Some results of the implementation of the method in real work are shown. In conclusion, the determining parameter towards optimal control of preventive action is the direct and active participation of workers in safety matters.

## 1. Introduction

This research is based on the critical analysis of several risk assessment methodologies and their application in building projects, during which the need to establish the parameters which best reflect the reality of the construction work environment was detected, spanning four techniques for combating risk (Safety at Work, Industrial Hygiene, Ergonomics and Psychosociology), taking risk tolerance into account. Where risk tolerance is concerned, other considerations are: the factor of the reality of the construction sector, which includes the risks associated with the complexity of the units of work, their location and interdependence [1,2]; the factor of the contractor’s economic response in terms of the direct implementation of construction systems and prevention methods [3,4]; and the social factor, including the interest, participation and the worker emotional states [5,6]. These factors span the documentary environment [7,8], the construction environment [9,10,11,12], and the social environment as fundamental elements associated with the execution of a project [5,9,13,14,15].

The difficulty and characteristics of the construction project environment establish a directly proportional value of complexity which affects the initial guidelines set out in the documentary environment [13]. However, the development of preventive systems, social activity, roles, hierarchies, and work-related stress add evaluable parameters to the construction environment which are of a corrective or inversely proportional nature [16,17].

It is fundamental that the number of accidents on building sites is reduced to a reasonable value by applying accident prevention systems and managing the risks [6,18]. Nevertheless, risk assessment in construction projects requires a detailed study of the different risks to which the construction workers are exposed [12,19]. As a basic criterion, usually there are three phases required by assessment methodologies in order to establish a risk assessment [20,21]:Identification of the potential dangersAssessment and quantification of the riskCategorization

A parametric selection must be applied to these criteria for the purposes of subjective analysis (qualitative) and objective analysis (quantitative) when evaluating risk [5,17]. Although the evaluation criteria are usually individually aimed at each technique for combating risk [22], processes in the construction industry present the combined complexity of management, organization (interdependence between units of work, different project execution phases…), environmental planning (climate, auxiliary resources…), human resources (staff, tasks, skills, temporality…), materials, etc. [18,23,24,25].

There are risk assessment methodologies which have extended the observation parameters in the search for their suitability of application for building projects [17]. Asilian considered it necessary to investigate the fundamental causes of the unsafe behaviors of workers in the construction sector thus creating a new effective tool [2]. As such, Forteza et al. [12,22] in their Global tool for assessing construction projects methodology, collect information about the structure of the project, the environment, the physical development, the agents and the type of project; in order to identify and assess the risks, the obstacles and the resources. Furthermore, Pinto [23] bases his QRAM methodology on the event tree by means of which certain risks that are characteristic of construction projects are analyzed based on four dimensions of observation: safety climate, severity factors (consequences), possibility factors (probability) and safety barriers (safety measures). Added to his formula is the fuzzy parameter based on the fuzzy sets theory [26]. Meanwhile, Reyes et al. [27] unifies criteria in the decision-making hierarchy and assesses the consequences of the variations which can be considered in relation to the construction lifecycle; from the design phase to renovation. The risk study that Sousa et al. [28] propose analyzes the accidents in construction industry projects, and they maintain that a correct project definition with adequate measures in the planning and organization of the project, can minimize accidents by more than 60%, emphasizing the need to manage health and safety during the construction lifecycle: design, execution, and use. Finally, Salanova et al. [29] establish the importance of studying psychosocial and ergonomic factors, since generally, risk assessments in construction consider physical, technical and management aspects, excessive confidence, and strained postures being the main causes of construction accidents. All these considerations imply the need for a culture of safety both in the construction project environment and in the construction process environment [17,18]. 

Accidents at work in the construction industry require risk analysis from an overall environment point of view [30,31,32,33,34,35]. Assessment methodologies, both those designed for the industry sector and those ultimately designed for the construction sector, establish observation and analysis criteria depending on the characteristics of the job roles and the particular focus of the methodologies themselves be it the identification of risks, the assessment and quantification of risk, or the classification of risk. Each and every methodology applies the parameters and protocol systems which best suit the objective sought, spanning, to a greater or lesser extent, the four techniques four combating risk: Safety at Work, Industrial Hygiene, Ergonomics, and Psychosociology. 

Different occupational risk assessment methodologies and their applicability on construction sites have been investigated, concluding that the results are very disparate depending on the type of method used. This is mainly due to the changing conditions during the execution of construction. To do this, a new method of occupational risk assessment is proposed adapted to construction sites and based on the assessment methods (Methodology Section) that have the greatest affinity in their implementation in construction sites [21,36,37].

The objective of this study is to establish the parameters and the protocol for the new occupational risk assessment methodology adapted for construction projects, called Level of Preventive Action. In addition, results of the implementation of the method and their effectiveness for the improvement of health and safety conditions in construction sites are attached.

## 2. Methodology 

The bibliography that raises the definition of risk (R) and the parameters of which it is composed is very extensive. In totality, the direct relationship between the probability (P) of an accident occurring and the consequences (C) of the same is accepted; as the product of the probability and the consequences (R = P · C) [16,37,38,39]. However, there is research that proposes another corrective parameter regarding exposure (E) to risk; with the expression R = P · C · E [36,39]. Finally, William T. Fine proposes to incorporate another correction parameter based on the degree of risk correction (G), being the expression R = P · C · (E/G), being an expression with a greater approximation to the real circumstances [39,40]. However, these parameters have a qualitative and quantitative definition that is very difficult to interpret and apply in construction works [21,36,37].

In total, there have been twenty-three risk assessment methodologies that have been investigated and analyzed from the point of view of applicability and usability in construction works is very broad [21,36,37]. It was determined that the evaluation methodologies analyzed very specific circumstances (safety risks, ergonomic risks, hygienic risks, or psychosocial risks), being necessary to apply different methodologies to be able to cover these four techniques to combat risk. The new risk assessment methodology, Preventive Action Level, assesses the four risk-fighting techniques, is a global method, and is adapted to construction processes [36]. This new methodology develops the same parameters as the William T. Fine methodology; adapting the parameters of degree of exposure and degree of correction to other new parameters from a new observation approach or preventive observation environment.

Based on the mathematical formula of William T. Fine’s methodology [40]; on the parameterization of the fundamental concepts identified in the INSST (National Institute of Workplace Health and Safety of Spain) assessment methods [41] using risk tolerance criteria; the ANACT method (French National Agency for the Improvement of Working Conditions) [38,42] using the importance of work criteria; the RNUR method (French National Renault Factories Authority) [43] using personal satisfaction criteria; the FINE method [40,44] using the incorporation of the justification of preventive action; and on the analysis of the documentary environment, (which reflects the complexity of the construction systems, their location and interdependence), the construction environment (referring to the worker’s exposure to the construction system and the individual and collective means of protection employed) and the social environment (with respect to the worker’s perception of the health and safety of the environment, and their emotional states) of a construction site [36]; the parameters of the new Level of Preventive Action risk assessment formula are formalized (with the acronym L_pac_) and the phases of action in the protocol are established:The first phase of the protocol defines a characteristic value of the risk inherent to the work situation being observed. Starting from the quantification of the qualitative parameters that define the probability and consequences of the risk.The second phase assesses the impact on the risk of the documentary environment, construction environment, and the social environment. Starting from the calculated characteristic value, said value will be required higher or lower depending on the incidence regarding the assessed risk.The third phase indicates the criteria for preventive action control, using the value obtained from the Level of Preventive Action (L_pac_) in relation to absolute risk (Ab_r_), as a deviation from the initial preventive action. This value indicates the amount of preventive action that is required to achieve an optimal control situation of preventive action during construction. The control of preventive action may be determined precisely in the assessed risk or, more generally, in each of the observation environments or, globally, with respect to the general situation of the construction.The fourth phase indicates the recommendations for action, based on the amount of preventive action that is required to be provided to achieve optimal control of the preventive action. Preventive actions may be established on the assessed risks, the preventive environments, or on the work in general.And in the fifth phase, the improvement of the preventive action during the work process is verified. Depending on the risks that are assessed and the immediate nature of the new risk assessment method, the improvements can be verified once the preventive actions have been implemented.

Based on the definition of the mathematical formula proposed by the new occupational risk assessment method, it has been implemented in a real construction site. Data has been taken on the observation environments (documentary, constructive, and social) in their corresponding parameters. The data collection is characterized by being a specialized technical observation in construction works and by a simple survey, which is carried out to the workers and agents of the work, of a psychosocial nature. The results obtained justify the validity of the theoretical contents of the method and determine the amount of preventive action on the risks evaluated.

Data has been taken from a real construction process located in the province of Madrid (Spain), for the construction of 6 semi-detached single-family homes, with three floors, one floor below ground level for garage and storage room, and two floors above ground level for housing. The total constructed area of the building is 1528.26 m^2^. The data presented correspond to the collection of data on-site from June 17, 2016, to April 27, 2017. We collected data weekly, for 34 working weeks and 73 items or construction systems inspected. The elevation plan of the project, a current view of the building, and various risk situations during the construction works are shown schematically (Figure 1).

### 2.1. Phases of the Investigation

Until the design and application of this new method, the research was developed in three phases:Preliminary investigation phase. It began in 2013, from the analysis of different risk assessment methodologies and their usability in construction works. The evaluation was carried out on three construction systems with a high incidence of accidents [10,13,17,18,19,25,32] and on the most characteristic risks in construction sites [6,8,10,11]. The evaluation was carried out from the written and photographic documentation of three real construction works, already completed at that time. The construction works that were worked on were a detached house (644.81 m^2^ and completed in 2013), the rehabilitation of the facades, roofs and pavement of a street (with a total area of 2379.71 m^2^ and completed in 2011), and a public school (2741.68 m^2^ and completed in 2010) [21,36,37]. The conclusions of this preliminary research work evidenced the enormous disparity regarding the results of the risk assessment for the different methodologies [21,36,37].Experimental design phase. A new mathematical formula developed from the method of William T. Fine [40,44] is proposed. New prevention environments are incorporated into the definition of risk that depends on the probability and consequences: the documentary environment, the constructive environment, and the social environment. These environments determine the degree of correction in terms of prevention, defining the new parameter of Evaluation of the Level of Preventive Action during the construction process [36].Implementation phase of the new risk assessment method adapted to construction works, Level of Preventive Action. Specialized technical data is collected and a simple psychosocial survey is carried out in a real construction site with which there is no contractual link (see Acknowledgments), to guarantee greater objectivity in data collection. The construction site was visited once or twice every two weeks, even twice on the same day. The visiting hours were random. It was documented photographically and the results and recommendation criteria for the control of preventive action were communicated to the responsible personnel. The average time for evaluating each construction system was three hours (data collection and office work).

### 2.2. Investigation Procedure

Table 1 shows the 73 construction systems evaluated during the 34 work inspections and the number of workers who were surveyed at each work inspection.

The action procedure for each inspection was as follows:Photography, analysis, and assessment of the risk situation from outside the construction.Photography, analysis, and assessment of the risk situation of each construction system that was being carried out.Photography, analysis, and assessment of the risk situation of the protection and prevention systems; both collectively and individually.Photography, analysis, and assessment of the interest of building workers and agents in occupational health and safety regarding their participation.Psychosocial survey on the perception of the emotional state and the perception of risks, of workers and building agents.The risks that the evaluator determines to be appropriate are evaluated.From the spreadsheet, the quantitative and graphical results of the parameters of the risk assessment and the Level of Preventive Action are obtained.We proceed to determine the degree of demand in preventive action.The characteristic values for the preventive action controls are obtained.The Preventive Action Level results are offered for each risk, for risk groups (Occupational Safety, Industrial Hygiene, Ergonomics, and Psychosociology), and for all the risks evaluated.The recommendations for the control of preventive action cover the documentary environment, the constructive environment, the social environment, and in all the parameters evaluated.The recommendations must establish communication strategies that generate a preventive culture and establish principles for improving the perception of risk with the participation of workers and building agents.

## 3. Parameters of the New Risk Assessment Method

The new risk assessment method establishes how much the level of prevention is deviating from the initial approach; in the Health and Safety at Work Plan, determining the level of prevention that needs to be incorporated to the implementation of the project in order to improve the design conditions, construction conditions and social relationship conditions. It is a method which has been adapted for the “special” complexity of construction projects and spans the assessment of all of the techniques for combating risk: Safety at Work, Industrial Hygiene, Ergonomics, and Psychosociology. The parameters which define the Level of Preventive Action (L_pac_) are:

Absolute environment (Ab_e_), (documentary basis of initial design and Implementation Plan drafting phase, covering the basic risk parameters):Probability (P): estimation of the risk tolerance in terms of the probability of an injury occurring.Consequences (C): estimation of the risk tolerance in terms of the expected consequences of the injury which could occur.Absolute Risk (Ab_r_): estimation that is performed at the start of project implementation using the Health and Safety at Work Plan drafted by the construction company. It is the absolute documentary environment and serves as the basis for comparison for the rest of the parameters.

Documentary environment (D_e_), (procurement phase with the construction company, covering the basic risk parameters and the physical and geometrical risk parameters):Relative Risk (R_r_): preventive parameter which interprets the complexity of the construction safety typical of the unit of work and which increases the value of the absolute risk.Borderline Risk (B_r_): preventive parameter which interprets the location of the unit of work and its impact on the environment, and which increases the value of the absolute risk.

Construction environment (C_e_), (phase where the Execution Plan is implemented, covering the basic risk parameters, the physical and geometrical risk parameters and the construction and human resource parameters):Degree of Exposure (E): parameter which assesses the amount of time used in completing the unit of work and whether workers are exposed to the risk several times during implementation of the work unit, increasing the value of the absolute risk.Economic Capacity (E_c_): parameter which assesses the amount of economic resources in constructive prevention systems; and which decreases the value of the absolute risk.

Social environment (S_e_), (phase where the Execution Plan is implemented, covering the basic risk parameters, the physical and geometrical risk parameters, the construction and human resource parameters, and the satisfaction and participation level parameters):Participatory Interest (P_i_): parameter which assesses the interest in the participation of the different workers and agents involved in a construction project, by obtaining their perception of health and safety; and which decreases the value of the absolute risk.Level of Satisfaction (L_s_): parameter which looks at general aspects of behavior, state of mind and human attitude which have a significant influence, or may have a significant influence on the generation of risks; and which decreases the value of the absolute risk.

Lifecycle environment (Lc_e_), (phase where the finished building is handed over to the new users, covering cohabitation, use and building maintenance parameters). This environment is based on sub-environments which do not form part of the construction process (environment of warranties and responsibilities associated with constructing a building; documentary, legal and economic environment; use and maintenance environment; and environment covering neighbor relations). Due to its complexity, it requires a detailed study, adapted for the use and maintenance circumstances within the building lifecycle.

Risk, in a construction project, is a parameter which depends on the probability of damage occurring; on the expected consequences which may arise; and on a corrective factor associated with documentary, construction, and social environments. This factor assesses the preventive level (of Safety at Work, Industrial Hygiene, Ergonomics and Psychosociology) in the environment of a construction project, the fundamental expression of which is (1):Risk = Probability · Consequences · Environment; R = P · C · E,(1)

Probability and consequences are parameters which define the risk of a situation. These parameters are established as part of the risk assessment detailed in the Health and Safety at Work Plan drawn up by the contractor. They are values which serve as a basis for determining the prevention systems throughout the construction of a building. The value will serve as the basis for comparing the rest of the parameters, which is why it takes on a conceptual value of absolute risk (2):P · C = Ab_r_,(2)

The *environment* (E) (3), in a construction project, is a context for correction which assesses the preventive action of the *documentary environment* (D_e_) (physical and geometrical building parameters), of the *construction environment* (C_e_) (building and human resources parameters) and of the *social environment* (S_e_) (participation and personal satisfaction parameters). The environment factor defines the *assessment of the preventive action* (A*_pac_*) (4): E = D_e_ · C_e_ · S_e_,(3)
E = A_pac_,(4)

As such, the risk is defined by the relationship between the *absolute risk* (Ab_r_) and the *preventive action assessment* (A_pac_); and it establishes by how much the level of prevention deviates from the initial approach, set out in the Health and Safety at Work Plan, thereby determining the prevention level that needs to be incorporated to the project implementation in order to improve the design conditions, construction conditions and social relationships. This factor is defined as the *Level of preventive action* (L_pac_) and is expressed (5):L_pac_ = Ab_r_ · A_pac_,(5)

The relationship between the parameters which define the *Level of preventive action* based upon the *absolute risk* and the *preventive action assessment* is (6):L_pac_ = (P·C)·(D_e_·C_e_·S_e_) = (P·C)·((R_r_·B_r_)·(E·(1/E_c_))·((1/P_i_)·(1/L_s_)) = (P · C) · ((R_r_ · B_r_ · E)/(E_c_ · P_i_ · L_s_)),(6)
where,

Absolute risk, Ab_r_ is (2)

Assessment of preventive action, A_pac_ is (7)
A_pac_ = (R_r_ · B_r_ · E)/(E_c_ · P_i_ · L_s_), (7)

## 4. Protocol

Etymologically, protocol is understood to mean the detailed sequence of a scientific, technical, or medical process of action. The Level of Preventive Action Protocol establishes the scientific and technical criteria for quantifying absolute risk and the assessment of preventive action.

### 4.1. Quantification of the Absolute Risk

The absolute risk depends on the probability and consequence parameters. These parameters are determined qualitatively, applying the INSST method [41], to estimate risk tolerance. The qualitative values for probability are low, medium and high; and the qualitative values for consequences are slightly damaging, damaging, and extremely damaging.

When expressing the level of preventive action, a product is developed from several parameters. This implies that none of the parameters must have a value of “0,” since this would cancel out the expression or provide an indeterminate value. In order to avoid this circumstance, it was decided that the quantification values should observe linearity in accordance with the expression (8): f(x) = 2x + 1,(8)

Similarly, the values that x takes have a direct relationship with the qualitative assessment performed for the probability and the consequences. For an assessment of low probability and slightly damaging consequences, the value assigned is x = 0. For an assessment of medium probability and damaging consequences, the value assigned is x = 1. And, for an assessment of high probability and extremely damaging consequences, the value assigned is x = 2. Such that the quantification results for the assessment of probability, following the aforementioned expression of linearity are those shown in Figure 2:

Low probability and slightly damaging consequence f(0) = 1Medium probability and damaging consequence f(1) = 3High probability and extremely damaging consequence f(2) = 5

The INSST methodology operates by using a matrix containing the probability and consequence parameters to obtain a table of qualitative results which define the risk tolerance (Table 2):

The same table is used to quantify the probability and consequences using the values obtained from the linear expression (8). Operating with the same criteria established by the INSST based on the product of the probability of an accident occurring multiplied by the severity of the consequences. The resulting values follow the expression of the product of both functions (9) (10):fp(x) · fc(x) = (2x + 1) · (2x + 1) = (2x · 2x) + 2x + 2x +1,(9)
f(p·c)(x) = 4x^2^ + 4x + 1,(10)

The resulting graphical expression, for the interval of values x = 0, x = 1 and x = 2, which quantifies the absolute risk parameter, is the Figure 3:

Likewise, using the risk tolerance table, a matrix of results is obtained which quantifies the risk tolerance equivalent (Table 3).

Based on the polynomial expression which defines the quantification of the absolute risk the values f(x) = 1, f(x) = 3, f(x) = 5, f(x) = 9, f(x) = 15 and f(x) = 25 are plotted on the graph; which quantify the qualitative parameter of risk tolerance (Figure 4), without scaling the X-axis:

By scaling the X-axis and plotting the quantitative values of risk tolerance (1, 3, 5, 9, 15, and 25) we obtain the graph which quantifies absolute risk (Figure 5). These values and the graph serve as the basis for determining the rest of the parameters for the level of preventive action.

This quantification procedure is carried out using the compulsory risk assessment set out in the Health and Safety at Work Plan.

### 4.2. Quantification of the Preventive Action Assessment

The protocol for action is based on observation, technical analysis, and on collecting data in relation to the techniques for combating safety, hygiene, and ergonomic risk; as well as a psychosocial on-site survey. It is composed of five fundamental phases which analyze the real situation observed in the data collected:The first phase of the protocol defines a characteristic value inherent to the work situation being observedThe second phase assesses the impact on the risk of the construction environment and the social environmentThe third phase indicates the criteria for preventive action control, using the value obtained from the Level of Preventive Action (L_pac_) in relation to absolute risk (Ab_r_), as a deviation from the initial preventive action (Figure 2)The fourth phase indicates the recommendations for actionThe fifth phase verifies the improvement of the preventive action during the work process

#### 4.2.1. First Phase: The Characteristic Value

The characteristic value is associated with the characteristics of the unit of work being implemented, in relation to which each of the Level of Preventive Action parameters will be assessed, based on criteria concerning the complexity of the unit of work, the position of the unit of work, the degree of exposure to risk, organizational procedures, participative interest, and the congruence of the perceived risk.

The observation criteria for data collection, with regard to each of the parameters, is divided into three concepts depending on the degree of risk observed: low degree of risk, medium degree of risk, and high degree of risk.

This criterion is extremely helpful for interpreting observations and for correctly establishing the characteristic value. Each degree of risk concept observed is assigned a dual qualitative value (easy, difficult; a little, a lot; less, more; 0, 1; etc.) that is easily quantifiable. The scale of values for the characteristic value follows the same criteria as the quantification of the absolute risk (1, 3, 5, 9, 15, and 25). Each criterion analyzed in relation to the unit of work assessed will have a prior degree of risk which is identified as low or high; and its associated characteristic value from lowest to highest.

The risk assessment factor contains parameters which increase or decrease the characteristic value. In turn, the assessment criteria rise from lowest to highest for the documentary environment, construction environment and social environment parameters. Consequently, directly proportional parameters will increase their result for high characteristic values, due to the greater complexity, greater borderline risk and higher degree of exposure. Inversely proportional parameters, for high characteristic values, will correct the situation with greater organizational procedures, greater participative interest, and a higher level of satisfaction. 

In the first phase of the methodology, the observable reality is quantified using the characteristic value. The range of values is 1, 3, 5, 9, 15, and 25 (Figure 6) and one is assigned to each parameter of the Level of Preventive Action formula. 

#### 4.2.2. Characteristic Value of Relative Risk

The Relative Risk (R_r_) is the preventive parameter which interprets the complexity of the construction safety typical of the unit of work and which increases the absolute risk value. The construction factors, together with the scale of values and its associated characteristic value, depend on the graphics required for the implementation of the unit of work, the reformulation needed for the implementation of the unit of work, the number of workers that there are or which are necessary to implement the unit of work, the qualification required to implement the unit of work, the height or depth (in absolute terms, to the ground) of the working plane required to implement the unit of work, the tools and machinery required or existing to implement the unit of work, the weight of the necessary material, and the manageability of the necessary material when it comes to moving and installing it during the implementation of the unit of work.

Once the characteristic values of each construction factor analyzed have been selected, the arithmetic average of all of the values is calculated and rounded to the nearest characteristic value. 

An example of application is considered (Table 4) concerning the unit of work to construct a bare brick façade on a second floor. The average of the selected characteristic values is 8.25, so the characteristic value for the relative risk of the unit of work analyzed is 9. The midpoint is rounded up to the value directly above.

#### 4.2.3. Characteristic Value of Borderline Risk

Borderline Risk (B_r_): preventive parameter that interprets the location of the unit of work and its impact due to the environment, and which increases the absolute risk value. Where this parameter is concerned, two points of view are analyzed, considering the possibility of a fall:The actual height between the working plane and ground level. This concept is independent of the quantity of work that exists. The height is measured according to its true size and the depth in absolute terms.Location of the worker in relation to the situation of potential risk. This concept depends on the theoretical distance between the worker or workers and the dangerous situation. It is worth highlighting that there are safe and unsafe areas on work platforms. The characteristic value identifies the intermediate area between the safe area and the unsafe area, which is the borderline area.

Six borderline levels at height have been identified and five borderline areas have been characterized on the associated work platforms: For a working plane height of up to 1.0 m, the borderline area is established at a distance of 100 cm from the risk, and from this point there are 25 cm to the safe area and 25 cm to the unsafe areaFor a working plane height of up to 3.0 m, the borderline area is established at a distance of 100 cm from the risk, and from this point there are 25 cm to the safe area and 25 cm to the unsafe areaFor a working plane height of up to 5.0 m, the borderline area is established at a distance of 125 cm from the risk, and from this point there are 25 cm to the safe area and 25 cm to the unsafe areaFor a working plane height of up to 9.0 m, the borderline area is established at a distance of 150 cm from the risk, and from this point there are 25 cm to the safe area and 25 cm to the unsafe area.For a working plane height of up to 15.0 m, the borderline area is established at a distance of 175 cm from the risk, and from this point there are 25 cm to the safe area and 25 cm to the unsafe areaFor a working plane height of more than 15.0 m, the borderline area is established at a distance of 200 cm from the risk, and from this point there are 25 cm to the safe area and 25 cm to the unsafe area

The diagram in Figure 7 shows the elevations and distances from the areas of risk.

The characteristic value is identified in the borderline area at every level from the ground. The values assigned range from 1 to 25 (in whole numbers). As such, the values have been distributed proportionally depending on the hazard posed by the height. 

An example has been set out for the construction of a brick façade on the first floor (Table 5). The working plane is less than 5.0m, since the height of the enclosure is more than a meter and a half and requires the support of scaffolding. The workers are located within the area of risk, since the enclosure is at the edge of the platform. The characteristic value is 6.

#### 4.2.4. Characteristic Value of the Degree of Exposure

Degree of Exposure (E): parameter that evaluates the amount of time that is used to complete the unit of work and during which the worker is consequently exposed to the risk several times during the execution of the unit of work. It is a value that increases the absolute risk value.

This parameter not only analyzes whether, through technical observation, the time that is being used to execute the unit of work is too little or too much, but also whether the worker is being exposed to risks when carrying out the work itself and whether the environment leading to the place where the unit of work is to be implemented poses further risks. The factors affecting the exposure to risk together with the scale of values and its associated characteristic value are:In relation to observation for low-intensity exposure: in the case of never, the worker is not exposed to a situation of risk at work and there are neither internal nor external risks on the way to the job; the characteristic value is 1. In the case of rare, the worker is exposed to risk once or twice at work during the period of observation, there are no risks in executing the unit of work but there are low risks on accessing the place of work or there are low risks in executing the unit of work and no risks on accessing the place of work; the characteristic value is 3.In relation to observation for medium intensity exposure: in the case of unusual, the worker is exposed to risk between two and five times at work during the period of observation, there are low risks in executing the unit of work and low risks on accessing the place of work; the characteristic value is 5. In the case of occasional, the worker is exposed to risk more than five times at work during the period of observation, there are high risks in executing the unit of work and low risks on accessing the place of work or there are low risks in executing the unit of work and high risks on accessing the place of work; the characteristic value is 9.In relation to observation for high-intensity exposure: in the case of frequent, the worker remains intermittently exposed to the risk during the period of observation, there are high risks in executing the unit of work and high risks on accessing the place of work; the characteristic value is 15. In the case of continual, the worker remains exposed to the risk during the execution of the unit of work, the risks during the execution of the unit of work are dangerous and the risks on accessing the place of work are dangerous; the characteristic value is 25.

The situation, which is closest to both concepts must be identified, which implies a characteristic value for each criterion. An arithmetic average is calculated for the selected values and this is rounded to the nearest characteristic value. The midpoint is rounded up to the value directly above. 

In the example provided of the construction of a bare brick façade on a first floor (Table 6), the worker was exposed to the risk of falling from a height, to the risk of falling on the same level, to thermal risk and to relationship risk, more than five times during the period of exposure, which implies a characteristic value of 9. Likewise, the working environment features high risks in terms of the hazards due to incorrect work organization; and the work access environment features high risks in terms of the risks of entrapment by or between objects and risks due to movement, which implies a characteristic value of 15. The arithmetic average of both values is 12, since this is a midpoint, it is rounded up to 15.

#### 4.2.5. Characteristic Value of the Economic Capacity

Economic Capacity (E_c_): parameter which evaluates the organizational procedure of the execution of the work and the amount of economic resources invested in construction risk prevention systems; and which decreases the absolute risk value.

This parameter analyzes the prior organization, by means of observing the general organization in place, analyzing from the point of view of the individual worker, of the team or work group, and of the site in general. Likewise, it analyzes the amount of individual protective equipment that the workers have and the quantity of collective protection systems that there are on the site. With regard to the organization criteria, the analysis looks at whether there are mistakes, whether there is a lack of coordination, at the order during the execution of the work, whether the work is coordinated or whether all of the tasks require accuracy. Where the individual security measures are concerned, the amount of protection equipment that the worker is wearing is analyzed both in terms of the specific needs of the unit of work and in terms of the general site obligations. Where the collective safety measures are concerned, observation and analysis focus on whether the preventive systems proposed for the site are low, medium or high; and, likewise, whether the associated risks can be high, medium, or low, or if they can be eliminated 100%. The organizational procedure and prevention system factors, together with the scale of value and its associated characteristic value are:

In observing individual, team, and site organization, analysis is performed to see whether there are mistakes which imply having to redo the units of work and exposing the worker to the risks of the unit of work once again. This situation implies uncontrolled economic excesses for the company. 

Analysis is carried out into whether there is an uncoordinated situation between the workers during the execution of the unit of work. This indicates that there is interrupted work and little organization, which in turn implies an increase in the exposure to risks. This will result in delays to project progress.Analysis is carried out into whether there is insufficient order during project execution which would imply that there is confusion between workers.Analysis is carried out into whether there is order during the execution of the unit of work. The risks decrease by ensuring that the company’s economic investment remains in line with the project planning.Analysis is carried out into whether there is coordination between the on-site workers. This implies a significant reduction in the risks, which is reflected in the execution time and in the company’s economic investment.Analysis is carried out into whether the tasks are being executed accurately, which generates less risk and improves both the project execution time and economic return.

When observing preventive systems, the amount of individual protective equipment that the worker is wearing is analyzed both in terms of the specific needs of the unit of work and in terms of general site obligations. Analysis is also carried out into whether the collective prevention systems imply that the associated risks are high, medium, or low, or whether they have been completely eliminated.

In the example provided (Table 7) of the construction of a bare brick façade on the first floor of a building, the characteristic value of the economic capacity is analyzed by observing the organizational procedure and the project’s prevention systems. As such, with regard to individual organization, it is observed that the worker is organized in the execution of the unit of work; the associated value is 9. Where project team organization is concerned, confusion is observed between some workers, which implies a lack of order and an associated value of 5. With regard to general project organization, considerable disorder is observed, although there is no evidence that there is a lack of coordination between the different crews; the associated value is 5. In relation to the preventive systems, it is observed that in some cases, the workers’ individual protective equipment contains safety boots, suitable clothing (but not overalls) and that not all of them contain gloves. As such, it is considered that there are two sets of individual protective clothing and the associated value is 3. Turning to the collective protection equipment, it is observed that the site is fenced, the scaffolding is not perimeter scaffolding, the scaffolding does not feature collective safeguards, there are no handrails, the site is not clean and site access is not controlled; the associated value is 5. The arithmetic average of the values observed is 5.40 and the rounded value is 5. The midpoint between the characteristic values is rounded up to the characteristic value that is directly above.

#### 4.2.6. Characteristic Value of Participative Interest

Participative Interest (P_i_): parameter which evaluates the interest in the participation of the different agents involved in a construction project by obtaining their perception of health and safety; and it decreases the absolute risk value. 

This parameter is based upon observation, on the workers’ comments, and on the conversations held with them. Analysis is performed on the information that the worker is proved to have concerning prevention, training courses they have attended, knowledge in relation to health and safety at work regulations, and whether they request or demand that they are provided with individual and collective protection material. The involvement and participation of the worker in prevention procedures, both individually for their own units of work, and collectively for the remaining units of work are analyzed below. The scale of values refers to the participation in the health and safety of the site; whether they barely participate, participate a little, participate a lot, participate significantly, or even, if they demand risk prevention procedures. Another aspect to be considered is the participation of the workers and the company in risk prevention beyond the site and whether they are concerned about risk prevention systems spanning the neighboring environment. Analysis is carried out into whether the workers apply risk prevention systems aimed at the area surrounding the construction site, for the benefit of the local residents. The scale of values ranges from not having risk prevention systems to controlling risks beyond the site. Such risk prevention systems may be applied outside the site, such as cleaning, watering, windscreens and screens to prevent dust from spreading; or on the building site such as welding screens, sound barriers, dust nets, etc.

In the example provided of the construction of a bare brick façade on the first floor of a building (Table 8), the information that workers transmit concerning risk prevention systems was observed. Conversations were held with them in which they talked about their health and safety training, such that a lot of information was gathered on the subject; the associated value is 9. Where individual and group participation is concerned, this may be looked at using a survey or through observation, noting whether the participation is none, low, medium, high or there is active involvement in it. For individual participation, the value is 5 and for group participation, it is 3. In relation to the external appearance of the site, it is observed that there are no preventive elements beyond the building site aimed at the exterior; the associated value is 1. The arithmetic average of the values observed is 3.50 and the rounded value is 3. The midpoint between the characteristic values is rounded up to the characteristic value directly above.

#### 4.2.7. Characteristic Value of the Level of Satisfaction

Level of Satisfaction (L_s_): Parameter which considers general aspects of behavior, state of mind, and human attitude which significantly influence, or may significantly influence the generation of risk; and which decreases the absolute risk value.

This parameter is obtained by means of an on-site survey completed by all of the workers who are contributing to the construction project. The questions cover personal perception, safety perception, and environment perception criteria. Since it is a survey, it is important that the workers can easily interpret the scale of values. The worker’s responses shall fall within the linear scale of 1, 2, 3, 4, 5, and 6. The office work involved in the assessment will include a translation to the scale of characteristic values. All of the responses will be supported, and with the aim of assisting the worker, they will be offered an initial level of response such as low, medium, and high. The second level of response will depend on the first, since the low level will be associated with values 1 and 2; the medium level will be associated with values 3 and 4; and the high level will be associated with values 5 and 6.

With regard to personal perception, two fundamental concepts are necessary in this area: stress and state of mind. Stress is a very common phenomenon in today’s society. A distinction is made between eustress or positive stress (optimum level of activation required to carry out activities) which performs the function of protecting the body, and distress, or negative stress (excessive or inadequate level of activation in the body), which provokes failure in a person. As such, the best healthy habit is the prevention of stress [45,46], since the emotional and mental burden that work generates has consequences such as burnout, unhappiness, depression, immunological diseases, cardiovascular diseases, stomach disorders, musculoskeletal disorders, absenteeism due to illness and inability to work [47]. 

Changing states of mind causes human behavior to alternate between eustress and distress due to tension and tiredness; affecting health, diet, the amount of time we sleep, physical exercise, and everyday activities [48]. The states of mind are calm energy (positive state) when a person is working and is focused, energetic, but calm and relaxed; calm tiredness (positive sensations), when experiencing sensations of tiredness, drowsiness, sleepiness; tense energy (not so negative), when the sensations are of energy, liveliness and vigor, with feelings of tension and anxiety; tense tired (negative state), which appears when our resources are exhausted: the fatigue mixes with nervousness, the tension or anxiety generates an unpleasant state. 

The performance curve has been divided into six values (Figure 8) from being apathetic 1, being bored 2, being motivated 3, being focused 4, being upbeat 5, and being excited 6. Sensations or behaviors associated with fatigue, being exhausted and burnout are adverse health effects, affecting individual and organizational well-being [47], which are outside the measuring range, it being clear the issues that they cause and that they are incompatible with the healthy completion of a job. The state of mind trend has also been schematized.

For the survey concerning personal perception, the worker is asked to indicate which state of mind they identify with the most at the time of taking the survey. Where the perception of the environment is concerned, the worker is asked to indicate the number which best expresses the category, from a little to a lot. There are two conversions to the scale of characteristic values (Figure 9). Responses concerning personal perception shall be required in the numbering format 1-2-3-4-5-6 and the conversions to the characteristic values will be 1-3-5-25-15-9. Responses concerning the perception of the environment shall be required in the numbering format 1-2-3-4-5-6 and the conversions to the characteristic values will be 1-3-5-9-15-25.

In order to establish the characteristic value of the perception of the environment and safety, the relationship between the worker’s responses and those of the observer is ascertained, which adds a new concept concerning the congruence of the safety and that of the environment. This is due to the fact that the worker’s point of view is not the same as that of the observer. Due to the relationship that exists between the excess of confidence and work-related accidents in the construction industry [49], the worker’s responses are combined with those of the observer, and the sum is obtained. Subsequently, the arithmetic average is determined and the value obtained is analyzed for its closeness to the unit (Figure 10). When the result is 1 this means that there is congruence between the answers provided by the worker and the observer, and the characteristic value which is obtained is the maximum: 25. 

Four questions are put to the worker concerning their personal perception, the range of possible responses being: bored-1, upbeat-2, motivated-3, focused-4, challenged-5, and excited-6: What do you consider your state of mind to be?What would you say is your current energy level?What is your level of personal satisfaction? (outside the construction environment)What is your level of job satisfaction? (within the construction environment).

Two questions are put to the worker concerning their perception of the environment, the range of possible responses being: zero-1, very slight-2, slight-3, notable-4, considerable-5, and high-6: What level of complexity or difficulty do you associate with the work that you are doing? andWhat is the level of danger associated with the work that you do?

Four questions are put to the worker concerning their perception of health and safety at work, the range of possible responses being: zero-1, very slight-2, slight-3, notable-4, considerable-5, and high-6: How high is your level of participation in your own safety?How high is your level of participation in your colleagues’ safety?To what level do you consider your individual protective equipment to be complete?To what level do you consider the collective protection systems to be complete?

Three quantified responses are taken concerning personal perception, the perception of the environment and that of health and safety at work; the arithmetic average of the three is calculated. It is possible to obtain individual results, results by construction crew and results for all of the workers on the site. 

With regard to the example provided of the construction of a bare brick façade on the first floor of a building (Table 9), firstly the results of the survey are shown and secondly, the results of the characteristic value conversion are shown.

### 4.3. Second Phase: The impact on the Risk

Once the characteristic values have been obtained for each of the parameters in the assessment of preventive action, the impact of these characteristic values on the risks assessed is analyzed. In this study, the most characteristic risks for construction projects have been selected, in accordance with a guideline that is proportional to the classification set out by the Spanish Institute for Health and Safety at Work [44]:Safety, 010 Risk of people falling from a different heightSafety, 020 Risk of people falling from the same heightSafety, 040 Risk of objects falling during handlingSafety, 110 Risk of entrapment by or between objectsHygiene, 350 Risk due to thermal stressHygiene, 380 Risk due to inadequate lightingErgonomics, 420 Risk due to movementErgonomics, 440 Risk due to incorrect load handlingPsychosociology, 560 Risk due to personal relationshipsPsychosociology, 570 Risk due to incorrect work organization

For each of the risks to be assessed, analysis will be carried out to ascertain whether the characteristic value has a greater or lesser impact. As such there are two concepts which define the project circumstances: the characteristic value of the parameter to be observed, which serves as the basis for analysis; and the impact on the risk. This implies that the characteristic value may be higher or lower depending on the impact on the risk being assessed. The maximum and minimum values for each characteristic value are indicated (Figure 11). The study of the risks is detailed in each case and for each of the parameters in the assessment of the preventive action. 

Returning to the example of the construction of a bare brick façade on the first floor of a building, Table 9 shows the results of assessing each parameter and its impact on the risk. The absolute risk values are reflected based on the data in the Health and Safety Plan (drafted by the construction company during the procurement phase), and alongside them are the results of the characteristic values calculated for each assessment parameter. Table 10 provides the analysis of the impact of each parameter on each of the risks assessed and the results of applying the level of preventive action formulas (5) and (6).

### 4.4. Third Phase: Basis for Controlling Preventive Action

The level of preventive action results is used to establish the control criteria required to achieve the optimal prevention situation. These criteria are established through the quantification of risk tolerance (Figure 5. Graph showing the quantification of absolute risk). This implies that for each risk tolerance level (trivial, tolerable, moderate-low, moderate-high, significant, intolerable) there are six levels of preventive action control with a distribution that is directly proportional to the risk quantification formula, for each of the characteristic values:Optimal control of preventive actionAdequate control of preventive actionMore control of preventive actionGreater control of preventive actionIntense control of preventive actionExhaustive control of preventive action

Once the level of preventive action values has been obtained, the criteria are established for determining the degree of exigency for the assessment. In order to do this, the risk quantification graph is multiplied by a scale factor “ε”. This graph with the scale factor is expressed by the polynomial function: f(x) = (4x^2^ + 4x +1) · ε (11)

As a comparative table (Table 11), the levels of control for different levels of control exigency are expressed. The assessor chooses the scale factor and will depend on the results of the Level of Preventive Action. This implies that the scale factor should be adjusted to the highest of the results obtained for the Level of Preventive Action, when assessing each of the risks being analyzed and which form part of the unit of work.

Everything depends on the information contained in the documentation, on the observation of the structural circumstances, and on the social survey. 

For the example of the construction of a bare brick façade on the first floor of a building, a scale factor of ε = 4 was selected (12). As such, all of the values obtained will be compared with the graph obtained using the polynomial function (11): f(x) = (8x^2^ + 8x + 4)(12)

A graph is obtained (Figure 12) using the results which compare the value provided for the amount that the level of preventive action deviates from the assessment in the Project Health and Safety Plan (Absolute Environment), with the graph that results from applying the scale factor; and which determines, in the same risk quantification proportion, the values which identify the basis for controlling preventive action. 

Finally, the results are reported for the level of preventive action for each of the risks analyzed, and the amount of control required for each of the techniques for combating risk (Table 12).

### 4.5. Fourth Phase: Recommendation Criteria

The recommendation criteria are based on the percentages obtained. The recommendations can focus on a single risk, on a set of risks, on a single technique for combating risk or on all of the risks as a whole. Using the information obtained from the data collection and the surveys, specific recommendations can be made for each of the parameters in the level of preventive action formula. These recommendations cover the documentary environment, the construction environment, and the social environment. Likewise, particular recommendations may be established concerning a single type of risk, a set of risks, a technique for combating risk, or all of the risks as a whole. In order to achieve better results or optimal control responses, it is necessary to coordinate the preventive action by using the prevention levels in the risk assessment contained in the Health and Safety Plan; reaching an agreement with the planners on structural solutions which improve site safety conditions; reaching an agreement with the project agents on the points for improvement in the execution of the project and reaching an agreement with all of the people who make up the on-site team on psychosocial recommendations.

### 4.6. Fifth phase: Verification of Project Results Progress

This risk assessment methodology makes it possible to make recommendations from the very first site inspection that is carried out. The results are immediate and as such, it is possible to check the progress of the prevention systems, immediately, once the preventive action criteria have been implemented based on the amount of preventive action control to ensure an optimal control situation. This check can be determined on the risks evaluated in the different construction systems, in the different preventive observation environments and globally in the construction process of the work. 

## 5. Results and Discussion. Participation

This research is based on a theoretical analysis carried out on three very different types of construction: the construction of a single-family house, a rehabilitation of a street with the facades of several buildings, and the construction of a current building for school use. In this case, these constructions had already been completed and the evaluation was made based on the photographic information and the documentation of the inspections carried out during the works. On the techniques of fighting against the risk of Safety at Work, Industrial Hygiene, Ergonomics, and Psychosociology, an initial theoretical base of preventive action was established [21,37].

In the data collection carried out in real and real-time construction, a protocol adapted to the circumstances and the characteristic complexity of a construction site began to be implemented [36].

The method requires a relatively intense data collection to guarantee an adequate result of the Preventive Action Level. However, this data collection is carried out with a specialized technical observation in which objectivity prevails (Appendix A, Figure A1) and from a simple psychosocial survey on the perception of risk (Appendix B, Figure A2). These illustrations show two documents with information taken for an inspected construction system, from a spreadsheet. During the construction phase, 34 inspections were carried out, with a total of 73 construction systems inspected with technical data collection and psychosocial survey.

Next, different graphs of results are expressed. Figure 13 shows the global Level of Preventive Action (on a logarithmic base 10 scale) that is required in each of the construction systems and the safety perception of the workers’ environment and the evaluator environment. The graph shows the different levels of preventive action control: L_cap_ > 60% exhaustive control, L_cap_ > 36% intensive control, L_cap_ > 20% greater control, L_cap_ > 12% more control, L_cap_ > 4% adequate control and L_cap_ < 4% optimal control. It can be verified that exhaustive levels of control are required for the construction systems that imply greater risk in accidents: foundations, structures, floors, façades, and roofs [39,50,51]. The risk perception values for the workers were between 10 and 15. However, for the evaluator, the risk perception values were low (between 1 and 6). The perception of safety for the workers was higher than for the evaluator. However, intensive levels of control have been required in almost all construction systems, with safety and health conditions in construction being quite low.

Figure 14 shows the results of the Level of Preventive Action and the characteristic values for the Participatory Interest and Satisfaction Level parameters. The levels of participation in construction safety, both individually (one worker) and collectively (by crews and all workers on the site), were very low. During the construction of the foundations, the façades, the floors, and the roofs, the participation in safety was the lowest during the work. It can be seen how the levels of worker satisfaction were the lowest during the construction of the highest roofs of the building, with few protection systems against the risk of falling.

In Figure 15, the results of the global Level of Preventive Action for the entire work and the Preventive Action Level for the risk of falling at different heights are shown. It can be verified that this risk was present throughout the construction, requiring exhaustive levels of control for the construction systems of foundations, floors, façades, and roofs.

Based on the different levels of results that the method allows, it is possible to propose different preventive action controls. These preventive actions can be implemented on each of the evaluated risks, each of the evaluated construction systems, each work team, and each individual worker. Likewise, the approaches in the implementation of preventive actions can be oriented with respect to each of the techniques to combat risk: Safety at Work, Industrial Hygiene, Ergonomics, and Psychosociology. It is the responsibility of the Construction Health and Safety Coordinator during the construction works to communicate the preventive actions to the construction agents (promoter, designer, builder, and Facultative Direction) and to the workers.

During the process of collecting data, a lack of interest by all of the construction agents and workers in the subject of risk prevention was noted. The solution that is being considered is that someone needs to make safety decisions, and someone needs to take on the responsibility for prevention-related decisions. There is a significant lack of knowledge on this subject.

It is fundamental that all of the construction agents and workers participate in bringing about improvements to the prevention systems on construction sites. The Participatory Interest parameter (P_i_) is related to all of the formula parameters. The Absolute risk could decrease depending on the project planners’ participative interest in prevention, thereby improving the circumstances of risk probability and consequences. The documentary environment could improve depending on the project planners’ participative interest in prevention, thereby improving the physical and geometrical characteristics of the building. The construction environment could improve depending on the participative interest of the contractor, thereby improving the material, human, and prevention system resources of the project. In addition, the social environment could improve depending on the participative interest of all of the people who are working on the project, thereby improving participation and the level of satisfaction.

Construction conditions change very quickly, so data collection is relatively complicated due to the large number of technical items that must be verified in addition to the psychosocial survey. However, within the work of this research is the development of a computer application that facilitates such data entry. Once the spreadsheet has been created, the characteristic values of each of the parameters and the evaluation of each of the risks are obtained. The results offered by the method are immediate and determine the different levels of preventive action control.

The applicability of the method can be specific on a construction system, partial on a set of different construction systems or global on the entire construction work. Being able to evaluate a single risk or different types of risks. It is a method adapted to the own complexity that characterizes construction works. Being one of the peculiarities of the method its enormous sensitivity to detect risk situations. Figure 16 shows the results of the Preventive Action Level in its natural scale and it is verified that the peaks with the highest level of preventive action are identified with the construction systems with the highest incidence rate in accidents.

## 6. Conclusions

The action protocol is based on technical observation and data collection in relation to the safety, hygiene and ergonomics environments, and a psychosocial on-site survey. The most efficient and fundamental corrective parameter for optimal control is the Participatory Interest of workers in risk prevention.

This risk assessment methodology can be implemented on-site with the added advantage of enabling immediate assessment, such that it reflects the real and complex situation of the construction process, in the chain of command, with the aim of establishing the preventive control levels required to achieve the optimal prevention situation; thereby obtaining essential advance preventive control. 

A change in direction in the risk assessment systems used in construction projects is essential. The systematic evolution of assessment processes is necessary so as to ensure that they are adapted in an effective and practical manner. In this evolutionary context, the new risk assessment method (Preventive Action Level) is more flexible in its applicability and more sensitive so as to highlight the risks in all of the scenarios of the construction process, including four of the techniques for combating risk (Safety at Work, Industrial Hygiene, Ergonomics, and Psychosociology), and it manages to establish a relationship between the different levels of preventive action and the perception of risk that the workers have.

Two new parameters have been established in data collection; On the one hand, the Characteristic Value inherent to the observation environments with respect to the design, geometry, the material constitution of the building, the degree of exposure to risk of the workers, the number of prevention systems used by the construction company, to the amount of participation of workers in health and safety and the emotional states of workers; And on the other, regarding the degree of incongruity between the workers’ safety environment perception and the evaluator.

This new risk assessment methodology reflects the necessary conditions for evolution that the typical characteristics of a construction project require; and considers an alternative transformational approach, entailing new human concepts, emphasizing the participation, education, and awareness of both workers and construction agents in the improvement of the health and safety conditions job quality.

Observing people is fundamental; as a psychosocial preventive element; in each of the project environments (documentary, construction, and social) together with all of the elements encompassed by risk prevention in Safety at Work, in Industrial Hygiene and in Ergonomics. It is an evolutive act that is typical of human necessity.

The protocol for this new method of occupational risk assessment has a configuration adapted to the conditions of the construction process. It has been considered to establish the same category, in the characteristic values, to the new parameters. This implies a balance and better interpretation of the treatment of the results.

The applicability of the method can be partial or global. It is a method that allows the timely evaluation of any of the existing risks in the construction process. Additionally, globally on a set of risks and encompassing risk-fighting techniques in Occupational Safety, Industrial Hygiene, Ergonomics, and Psychosociology.

The great variety of results makes it possible to propose control recommendations for preventive action in a specific manner on a particular risk, on a particular worker, on a group of workers or work team, on preventive observation environments (absolute, documentary, constructive and social); or globally on the Level of Preventive Action in each of the techniques to combat risk or on construction in general.

It is essential, in the documentary environment, to identify and improve the previous preventive conditions regarding the geometry and constructive characteristics of the building, in the conception and design phase. In the construction environment, preventive action must improve the conditions of exposure to risk for workers and the collective and individual protection systems during construction work. Finally, participation in prevention and workers’ emotional states during construction work must be improved with positive communication strategies in the social environment.

This new methodology allows socializing with the different hierarchical levels of workers and building agents. The different levels of preventive action control will determine the degree of information and communication that will need to be implemented during the construction phase to achieve improvements in occupational health and safety. This has major implications for social relations between workers. It is important to highlight that, from the necessary preventive information, a necessary preventive training will be produced and, consequently, an improvement in the participation in the prevention of workers. This participation can be measured in the parameter of participatory interest, which will imply the improvement of the parameters of the documentary, constructive and social environments of the construction process.

## Figures and Tables

**Figure 1 ijerph-17-06369-f001:**
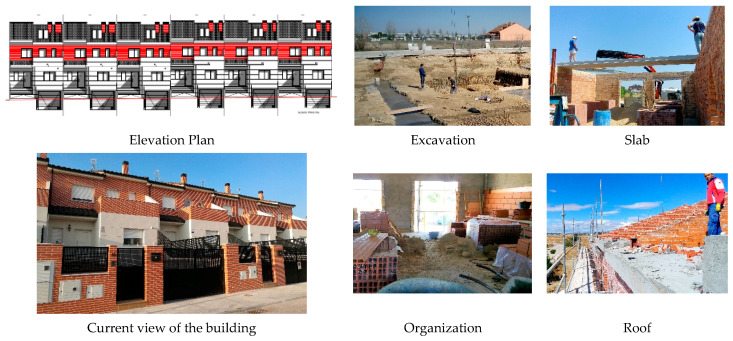
Elevation plan. Current view. Risk situations on site.

**Figure 2 ijerph-17-06369-f002:**
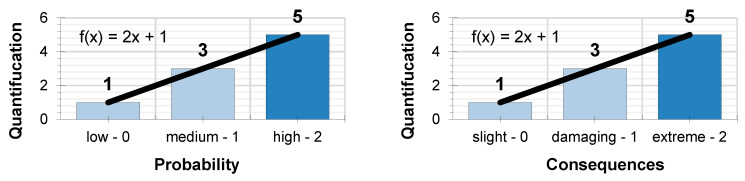
Quantification of the probability and consequences.

**Figure 3 ijerph-17-06369-f003:**
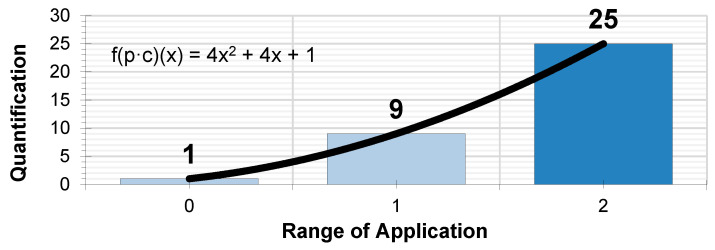
Quantification of probability and consequences.

**Figure 4 ijerph-17-06369-f004:**
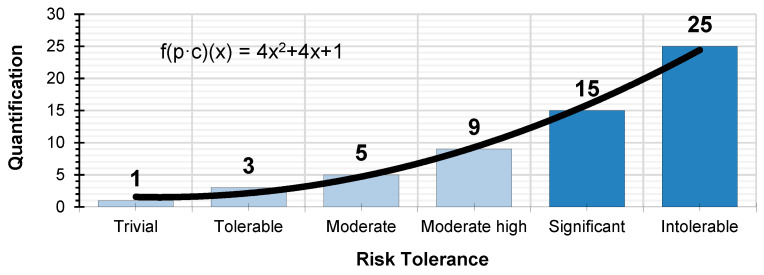
Quantification of risk tolerance.

**Figure 5 ijerph-17-06369-f005:**
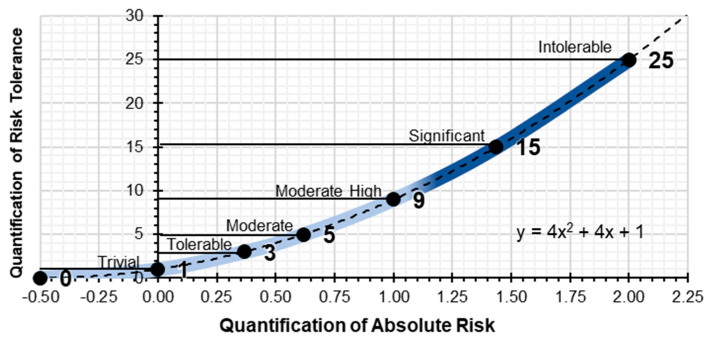
Graph showing the quantification of absolute risk.

**Figure 6 ijerph-17-06369-f006:**
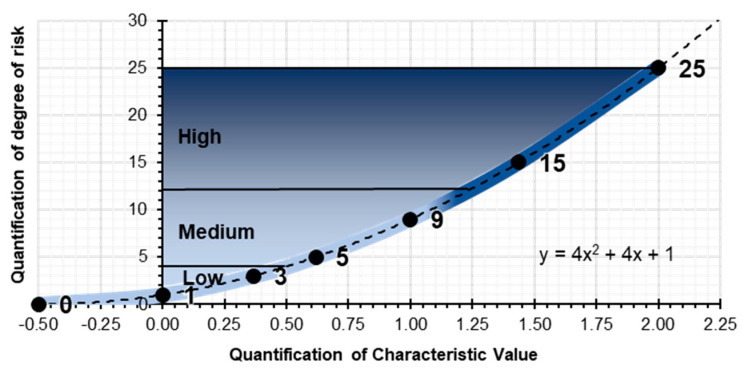
Graph showing the characteristic values.

**Figure 7 ijerph-17-06369-f007:**
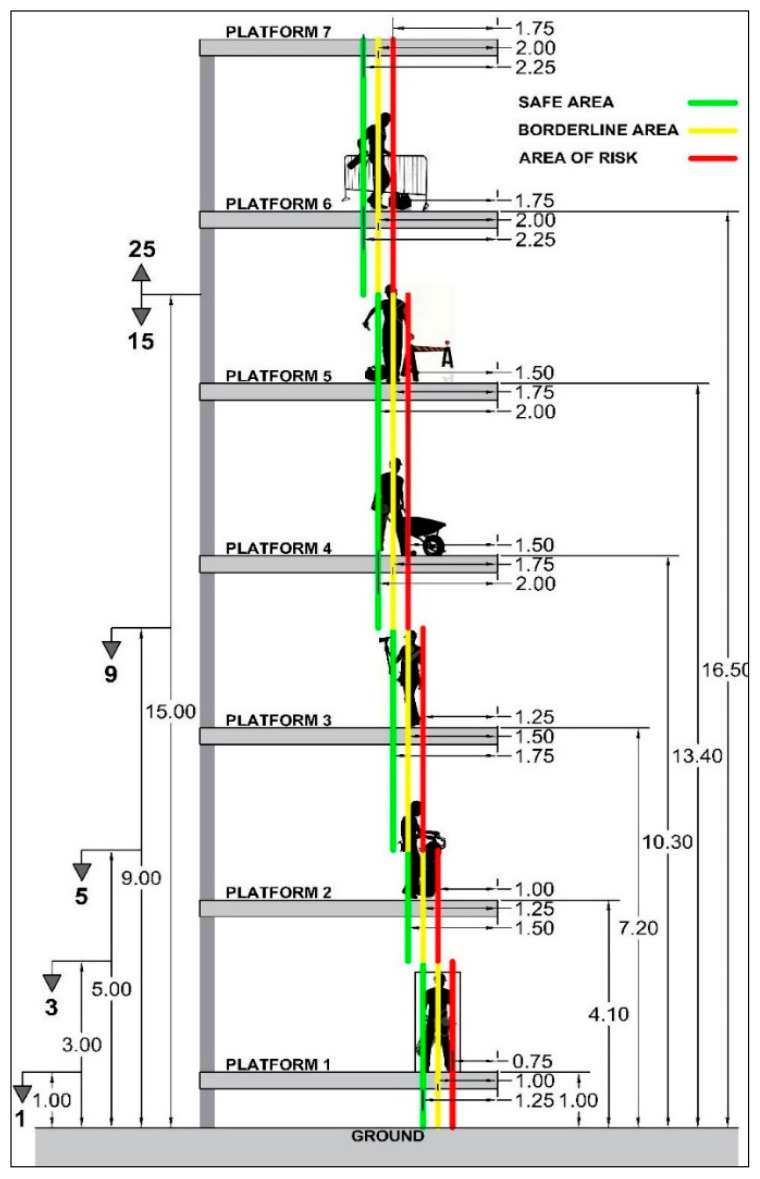
Diagram showing borderline risk.

**Figure 8 ijerph-17-06369-f008:**
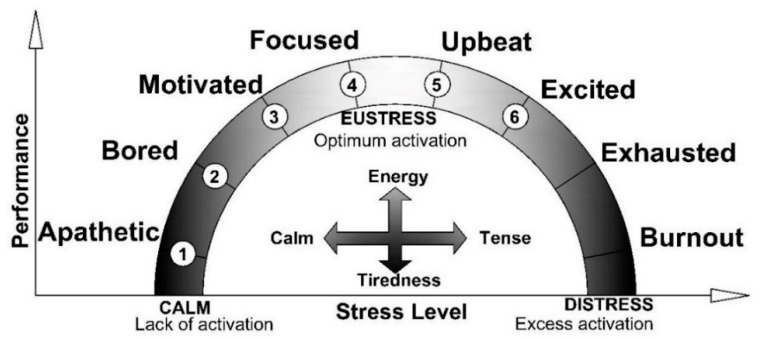
Performance curve.

**Figure 9 ijerph-17-06369-f009:**
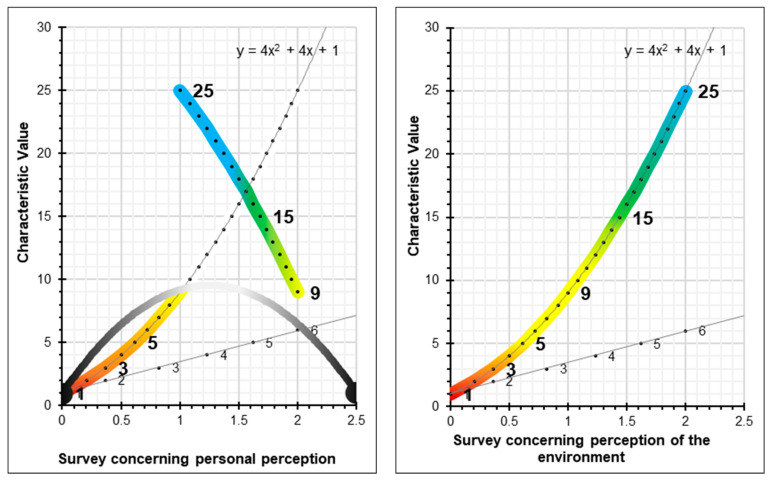
Graphical representation of responses using the characteristic values.

**Figure 10 ijerph-17-06369-f010:**
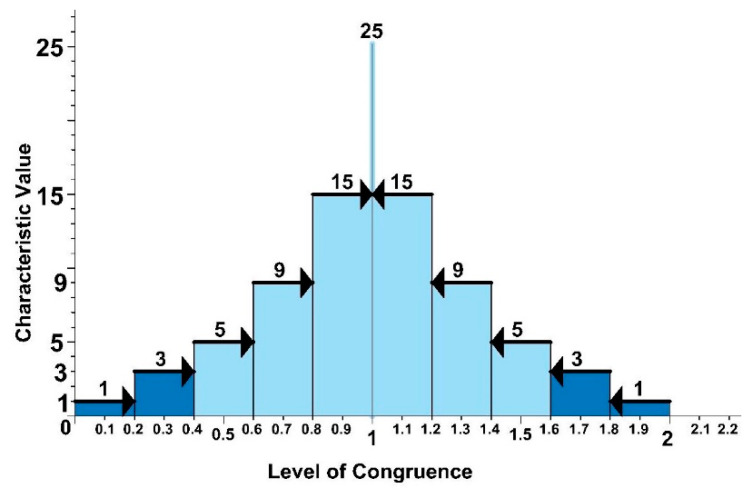
Congruence of safety and environment.

**Figure 11 ijerph-17-06369-f011:**
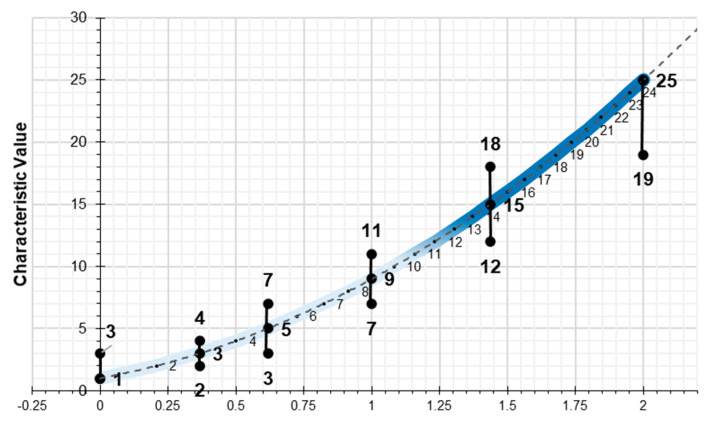
Variation in maximum and minimum for each characteristic value.

**Figure 12 ijerph-17-06369-f012:**
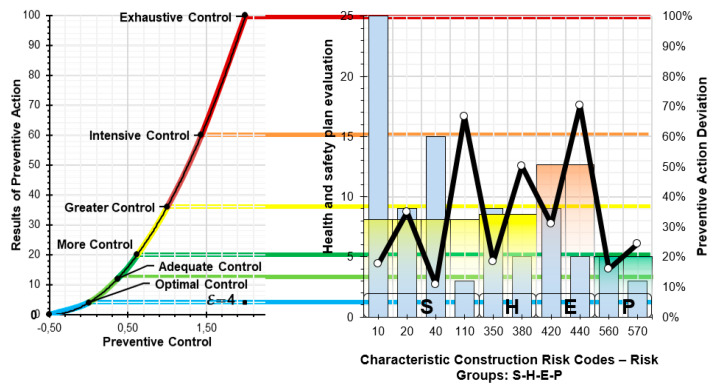
Results and comparative graphs.

**Figure 13 ijerph-17-06369-f013:**
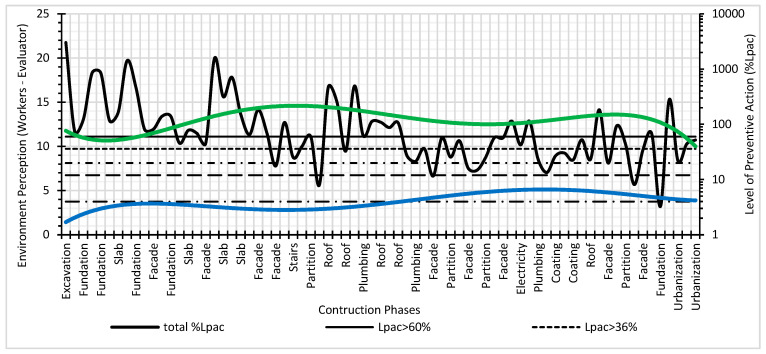
Level of preventive action and perception of the risk of the workers and the evaluator.

**Figure 14 ijerph-17-06369-f014:**
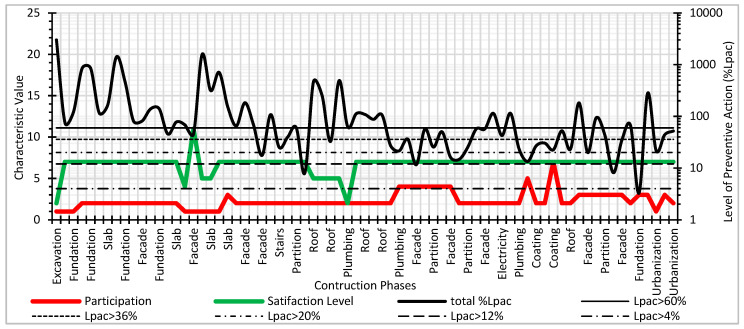
Level of Preventive Action and Parameters of Participation and satisfaction of workers.

**Figure 15 ijerph-17-06369-f015:**
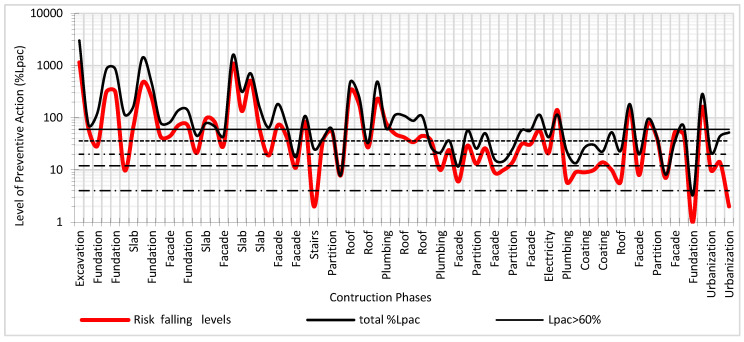
Level of preventive action for the risk of falling at different level.

**Figure 16 ijerph-17-06369-f016:**
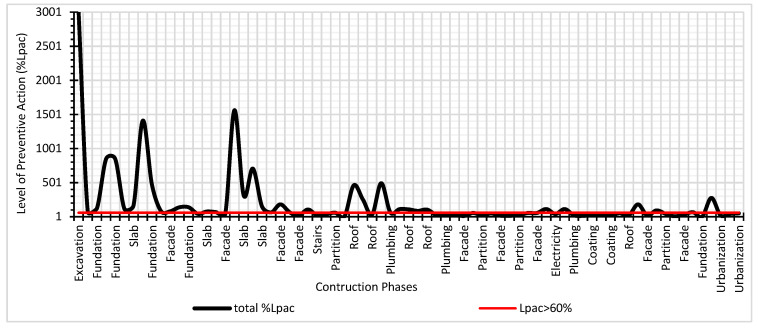
Global results of the Level of Preventive Action.

**Table 1 ijerph-17-06369-t001:** Chronology of the data collection procedure.

Number of Inspection	Date	Num.	Constructive Systems	Workers Surveyed
1	2016-06-17	1	Foundations	9
2	2016-06-23	2	Sewer inst.	9
		3	Plumbing Inst.	
		4	Foundations	
3	2016-07-01	5	Foundations	10
		6	Slab	
4	2016-07-06	7	Facade	8
		8	Foundations	
5	2016-07-12	9	Foundations	10
		10	Urbanization	
6	2016-07-14	11	Facade	14
		12	Foundations	
7	2016-07-26	13	Foundations	9
		14	Foundations	
8	2016-08-01	15	Slab	8
9	2016-08-10	16	Facade	5
10	2016-08-19	17	Facade	8
11	2016-08-26	18	Slab	6
		19	Facade	
12	2016-09-06	20	Slab	5
13	2016-09-14	21	Facade	5
14	2016-09-22	22	Slab	8
		23	Facade	
		24	Partitions	
15	2016-10-06	25	Stairs	8
		26	Facade	
		27	Stairs	
16	2016-10-14	28	Partitions	7
		29	Partitions	
17	2016-10-18	30	Partitions	4
18	2016-10-26	31	Roof	4
19	2016-11-08	32	Roof	7
		33	Partitions	
20	2016-11-17	34	Roof	7
		35	Plumbing Inst.	
21	2016-11-21	36	Roof	4
		37	Roof	
22	2016-12-16	38	Roof	11
		39	Roof	
		40	Plumbing Inst.	
23	2017-01-04	41	Coating	12
		42	Partitions	
		43	Facade	
24	2017-02-12	44	Coating	7
		45	Partitions	
25	2017-01-19	46	Facade	10
		47	Coating	
		48	Urbanization	
26	2017-01-25	49	Flooring	8
		50	Partitions	
27	2017-02-03	51	Flooring	7
		52	Facade	
28	2017-02-17	53	Flooring	10
		54	Electricity Inst	
29	2017-02-24	55	Urbanization	13
		56	Plumbing Inst.	
		57	Coating	
		58	Coating	
		59	Coating	
30	2017-03-02	60	Coating	6
		61	Flooring	
		62	Roof	
31	2017-03-16	63	Flooring	12
		64	Facade	
32	2017-03-30	65	Partitions	11
		66	Coating	
		67	Facade	
		68	Foundations	
33	2017-04-10	69	Coating	11
		70	Foundations	
34	2017-04-27	71	Urbanization	6
		72	Coating	
		73	Urbanization	

**Table 2 ijerph-17-06369-t002:** Risk estimation qualitative values

Risk Estimation		Severity of the Consequences		
		Slightly Damaging	Damaging	Extremely Damaging
**Probability**	**Low**	Trivial risk	Tolerable risk	Moderate risk
	**Medium**	Tolerable risk	Moderate risk	Significant risk
	**High**	Moderate risk	Significant risk	Intolerable risk

**Table 3 ijerph-17-06369-t003:** Risk estimation. Qualitative values.

Risk Estimation			Severity of the Consequences					
			Slightly Damaging		Damaging		Extremely Damaging	
			1		3		5	
**Probability**	**Low**	**1**	Trivial	1	Tolerable	3	Moderate low	5
	**Medium**	**3**	Tolerable	3	Moderate high	9	Significant	15
	**High**	**5**	Moderate low	5	Significant	15	Intolerable	25

**Table 4 ijerph-17-06369-t004:** Example of characteristic value in relative risk.

Relative Risk–Complexity of the Unit of Work Observed								
**Construction Factor**	Low Complexity		Medium		High Complexity		Characteristic Value	
	1	3	5	9	15	25		
**Graphics**	Without graphics	Verbal indication	Sketch	Technical detail	Formal plan	Technical equipment	9	Average:
**Setting-out**	Without reformulation	Reformulation by workman	Reformulation by supervisor	Reformulation by project manager	Technical equipment	Technical precision	5	8.25
**Workers**	1	2	3	4	5	≥6	9	
**Qualification**	Laborer	Foreman	Supervisor	Manager	Project Manager	Expert	5	
**Auxiliary Systems Level**	Ground level	≤1 m	≤3 m	≤5 m	≤9 m	>9 m	15	Total:
**Tools, Machinery**	None	Handheld tool	Hand-operated machine	Average vehicle	Heavy vehicle	HGV	15	9
**Material Weight**	Light	≤9 kg	≤50 kg	≤150 kg	≤500 kg	>500 kg	3	
**Manageability**	1 person	2 people	3 people	With a machine	With a crane	Complex system	5	

**Table 5 ijerph-17-06369-t005:** Example of characteristic value in borderline risk.

Borderline Risk–Position of the Unit of Work Under Observation															
Place	Specification	Height from the Ground												Characteristic Value	
		<1 m		<3 m		<5 m		<9 m		<15 m		>15 m			
**Safe Area**	**Within Safe Area (S)**	>125	1	>125	2	>150	4	>175	7	>200	12	>225	19	Height	6
**Borderline Area**	**Close to Borderline** **Area (BA)**	125	1	125	2	150	4	175	8	200	14	225	21	5	
	**Borderline (B)**	100	1	100	3	125	5	150	9	175	15	200	22	Area	6
	**Close to Area** **of Risk (AR)**	75	2	75	4	100	6	125	10	150	16	175	23	R	
**Area of Risk**	**Within Area** **of Risk (R)**	<75	2	<75	4	<100	6	<125	11	<150	18	<175	25	6	

**Table 6 ijerph-17-06369-t006:** Example of characteristic value in the degree of exposure.

Degree of Exposure to Risk						
Intensity of the Exposure		Number of Times that Workers are Exposed to the Risk During the Observation Period		Characteristic Value		
		During the Execution of the Unit of Work	In the Work Environment and Accesses to It			
**High**	**Continual**	The worker remains exposed to the risk	Dangerous internal and/or external risks	25	25	12
	**Frequent**	The worker remains intermittently exposed	There is a high internal risk and high external risks	15	15	
**Medium**	**Occasional**	The worker is exposed to the risk more than 5 times during observation	There are high internal risks and/or low external risks	9	9	15
	**Unusual**	The worker is exposed to the risk between 2 and 5 times during observation	There is a low internal riskand low external risk	5	5	
**Low**	**Rare**	The worker is exposed to the risk 1 or 2 times during observation	There is no internal risk and/or there is a low external risk	3	3	
	**Never**	The worker is not exposed to the situation of risk	There are neither internal nor external risks	1	1	

**Table 7 ijerph-17-06369-t007:** Example of characteristic value in economic capacity.

Economic Capacity-Organizational Procedure for Work Execution and Safety								
Procedures	Low		Medium		High		Characteristic Value	
	1	3	5	9	15	25		
**Individual Organization**	Significant error	Highly disorganized	Little order	Order	Coordinated	Accurate	9	5.40
**Team Organization**	Errors	Disorganized	Little order	Order	Coordinated	Accurate	5	
**Work Organization**	Errors	Disorganized	Little order	Order	Coordinated	Accurate	5	
**Individual Protection**	None	2 teams	4 teams	5 teams	6 teams	>6 teams	3	5
**Collective Safeguards**	None	Very high risk	High risk	Medium risk	Low risk	100% free of risk	5	

**Table 8 ijerph-17-06369-t008:** Example of characteristic value in Participative Interest.

Participative Interest–Participative Interest in Risk Prevention								
Procedure	Low		Medium		High		Characteristic Value	
	1	3	5	9	15	25		
**Worker Information**	No Information	Little Information	Adequate Information	A lot of Information	Significant Information	Demands Information	5	3.50
**Individual Participation**	Does not participate	Barely participates	Participates a little	Participates a lot	Participates significantly	Demands safety	5	
**Group Participation**	Do not participate	Barely participate	Participate a little	Participate a lot	Participate significantly	Demand safety	3	3
**External Appearance of Site**	No safety	Scarce safety	Little safety	Offers safety	Significant safety	Well maintained	1	

**Table 9 ijerph-17-06369-t009:** Example of characteristic value in the level of satisfaction. Survey and congruence.

**Level of Satisfaction—1. Survey Carried Out On-Site**																														
**Personal Perception**				**Perception of Risk**																						**Congruence**				
				**Perception of the Environment**								**Perception of Safety**																		
**Emotional States**	**Energy**	**Personal Satisfaction**	**Job Satisfaction**	**Difficulty of the unit of work**				**Danger of the environment of the unit of work**				**Individual participation in safety of unit of work**			**Group participation in site safety in general**				**Complete Individual Protective Equipment**				**Collective safeguards**			**Personal Perception**		**Worker’s perception of risk**	**Assessor’s perception of risk**	**Total**
				Laborer		Assessor		Laborer		Assessor		Laborer		Assessor	Laborer		Assessor		Laborer		Assessor		Laborer		Assessor					
6	6	6	6	1		1		3		5		6		25	5		15		6		25		5		15	9		0.1	0.3	13
9	9	9	9	4		9		5		15		1		1	2		3		1		1		1		1			1	3	
**Level of Satisfaction—2. Congruence of the Perception of Risk**																														
**Influence**			**Zero**	**Very slight**		**Slight**			**Notable**					**Considerable**							**High**							**Characteristic Value**		
**Personal Perception**			1	2	3	4	5	6	7	8	9	10	11	12	13	14	15	16	17	18	19	20	21	22	23	24	25	13	12.33	
**Congruence of Safety**			0			0.2		0.4		0.6		0.8		1			1.2		1.4		1.6		1.8		2			0.6		
**Perception of Safety**			1			3		5		9		15		25			15		9		5		3		1			9	15	
**Congruence of the Environment**			0			0.2		0.4		0.6		0.8		1			1.2		1.4		1.6		1.8		2			1.1		
**Perception of the Environment**			1			3		5		9		15		25			15		9		5		3		1			15		

**Table 10 ijerph-17-06369-t010:** Example of risk impact calculation.

Risks to Assess		Observation environments									Preventive Action	
		Absolute			Documentary		Construction		Social			
Code	Description	C	P	Ab_r_	R_r_	B_r_	E	E_c_	R_i_	L_s_	A_pac_	L_pac_
					Characteristic values							
					9	6	15	5	3	5	3.60	
**010**	**Different level**	5	5	25	8	7	15	4	4	12	4.38	18%
**020**	**Same level**	3	3	9	10	5	12	4	4	12	3.13	35%
**040**	**Handling**	3	5	15	9	5	12	6	4	14	1.61	11%
**110**	**Entrapment**	1	3	3	10	5	12	5	4	15	2.00	67%
**350**	**Thermal**	3	3	9	7	5	12	4	4	16	1.64	18%
**380**	**Lighting**	1	5	5	8	5	13	4	4	13	2.50	50%
**420**	**Movement**	3	3	9	10	5	16	6	4	12	2.78	31%
**440**	**Loads**	1	5	5	9	5	15	4	4	12	3.52	70%
**560**	**Relationships**	1	5	5	8	2	12	4	4	15	0.80	16%
**570**	**Organization**	1	3	3	10	2	14	6	4	16	0.73	24%

**Table 11 ijerph-17-06369-t011:** Scale factor.

Levels of Control	Exigency Factor for Controlling Preventive Action							
	Color	ε = 0.4	ε = 1	ε = 2	ε = 4	ε = 5.4	ε = 10	ε = 20
**Exhaustive control**		10.0	25.0	50.0	100.0	135.0	250.0	500.0
**Intensive control**		6.0	15.0	30.0	60.0	81.0	150.0	300.0
**Greater control**		3.6	9.0	18.0	36.0	49.0	90.0	180.0
**More control**		1.2	5.0	10.0	20.0	27.0	50.0	100.0
**Adequate control**		0.4	3.0	3.0	12.0	16.0	30.0	60.0
**Optimal control**		0.1	1.0	2.0	4.0	5.0	10.0	20.0

**Table 12 ijerph-17-06369-t012:** Results.

Code	Risk	Tolerance	A_pac_	L_pac_	Control	Technique	L_pac_
010	**Different level**	Intolerable	4.38	18%	More control	Safety	32%
020	**Same level**	Moderate high	3.13	35%	Intensive control		
040	**Handling**	Significant	1.61	11%	Adequate control		
110	**Entrapment**	Tolerable	2.00	67%	Exhaustive control		
350	**Thermal**	Moderate high	1.64	18%	More control	Hygiene	34%
380	**Lighting**	Moderate	2.50	50%	Intensive control		
420	**Movement**	Moderate high	2.78	31%	Greater control	Ergonomics	51%
440	**Loads**	Moderate	3.52	70%	Exhaustive control		
560	**Relationships**	Moderate	0.80	16%	More control	Psychology	20%
570	**Organization**	Tolerable	0.73	24%	Greater control		

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
