# Peer review of "Development of the Protocol of the Occupational Risk Assessment Method for Construction Works: Level of Preventive Action"

_ijerph, 2020, doi:10.3390/ijerph17176369_

Round 1
Reviewer 1 Report
See attached document

Reviewer 2 Report
The problem of describing risks in construction is not a new issue. However, an innovative approach to the construction environment element (E) should be emphasized, because the probability and consequences are found in most studies.The article lacks a clear and unambiguous definition of risk. It should be noted that the literature review carried out by the authors is poor. Consideration could be given to describing the social significance of the research more widely, especially in terms of Life Cycle Balance. The research was carried out correctly and the results achieved are very interesting and make a big contribution to the development of the field.
Round 2
Reviewer 1 Report
This is clearly a much better version of the paper. The additional text, figures and graphs definitely bring much more clarity overall to the authors' project. The authors have generally answered my comments and suggestions.
Some uncertainties remain, however. In their answer, the authors mention (or at least one of them does) mention the period of time during which data collection took place, but in the text the description of the methodology remains somewhat nebulous. Throughout the article there seems to have been in fact two data collections, one on real construction sites, and one which appears to have been a table-top exercise on finished projects. Some clarification would be required.
I am still convinced that the reference to employee participation is not just but one of the findings from the application of the risk assessment. From the beginning, it was never the main intention of this article, the main purpose being the development and testing of a new risk assessment methodology. "By the way", during data collection, they happened to notice that participatory interest was quite low; this study was not designed to demonstrate that it is a decisive factor. Such a demonstration would require a totally different research design. Therefore the authors should concentrate in this paper on the development and testing of their method, and, as they suggest in their response to my comments, devote a whole new paper on their finings from the data collection.
Provided that the reference to the decisive aspect of the participatory interest is removed from the title (as it is not the main purpose of the paper), and that a clearer step-by-step description of their research methodology (not the steps of their method, but the steps they as researchers took to develop the method and collect the data) is presented, this paper can now proceed to be published.
Author Response
Dear Reviewer
We are very grateful your comments and positively value the proposal to improve the final quality of the investigative work. The corrected items are already incorporated in the text.
In the attached file, your comments have been answered, justifying for the improvement of the article.
The texts of the answers are marked in green text.
In the main document, new modifications are shown in green text.
